# Laughter regulation in solitary and social contexts varies across emotion regulation strategies
Vanessa Mitschke ✉, Annika Ziereis, Sriranjani Manivasagam & Anne Schacht

Regulating amusement is crucial in social contexts where expressing amusement may be inappropriate or disruptive. Yet little research has directly compared the effectiveness of different strategies for laughter regulation. Across three experiments, we examined how distraction, cognitive reappraisal, and expressive suppression affect laughter-related facial expressions and amusement ratings during exposure to jokes. Laughter regulation was operationalized by means of facial electromyography (fEMG) and subjective ratings of funniness as proxies for the expression and experience of amusement. In Experiments 1 and 2 ($n = 40$ each), distraction and expressive suppression most strongly reduced facial activity, whereas reappraisal produced smaller but consistent effects. However, only reappraisal reliably decreased funniness ratings, suggesting selective effects on the cognitive evaluation of humor. Experiment 3 ($n = 41$) introduced social laughter feedback and revealed that the presence of another's laughter impaired expression control and increased funniness ratings, indicating that social cues shape both emotional expression and experience. Together, these findings show how distinct emotion-regulation strategies modulate amusement and laughter expressions in response to humorous stimuli and highlight the contextual sensitivity of laughter regulation in socially dynamic settings.

For more than two millennia, laughter has captivated philosophers, scientists, and artists alike, sparking debates about its meaning, origins, and role in human life. From Plato and Aristotle to modern theorists, laughter has been framed as everything from a marker of superiority to a release of psychic energy[1]. Extending this long-standing fascination, contemporary research emphasizes its many adaptive functions—enhancing well-being, strengthening social bonds, and regulating interpersonal dynamics[2,3]. Laughter is not a unitary behavior but a heterogeneous expressive response that can accompany a range of emotions—from joy and playfulness to embarrassment, schadenfreude, or derision[4]. In the present study, we focus specifically on amusement-related laughter, elicited by humorous stimuli and typically aligned with positive emotional meaning and affiliative intent. In this sense, laughter serves as the expressive manifestation of amusement, and its regulation provides a tractable means to study amusement regulation in everyday contexts. To study laughter regulation in a tractable and ethically viable way, we measured laughter-related facial muscle activity (using facial electromyography; fEMG) and subjective funniness ratings, which together serve as established proxies for amusement-related expressions and experiences during exposure to jokes. Yet laughter is not always welcome.

Most people can recall moments when they—or someone else—laughed at the wrong time, violating social norms or undermining the seriousness of a situation. These moments underscore the complexity of laughter regulation —the process by which individuals modulate the expression or experience of amusement to align with contextual demands. Regulating laughter, especially in contagious or emotionally charged contexts, can be as important— and as challenging—as expressing joy appropriately[5].

In situations where laughter is socially inappropriate, failure to regulate it can lead to serious interpersonal or reputational consequences. Laughing during solemn events (e.g., funerals, serious conversations, or disciplinary settings) may violate social norms, undermine the speaker's message, or be perceived as disrespectful or immature. Such moments can damage reputations, impair professional relationships, and trigger social sanction or embarrassment. For example, during a G7 meeting in 2019, former German Chancellor Angela Merkel appeared to suppress a laugh when U.S. President Donald Trump enthusiastically referenced his "German blood" while proposing a visit to Germany. The moment—captured on camera—circulated widely online and underscored the delicate balance that public figures must maintain between emotional authenticity and diplomatic decorum.

Department for Cognition, Emotion and Behavior, Institute of Psychology, University of Goettingen, Goettingen, Germany. ✉e-mail: vanessa.mitschke@uni-goettingen.de

Given that laughter is also highly contagious and often involuntary, regulating it—when necessary—becomes not only a matter of self-regulation, but a socially critical skill. In the present work, we therefore conceptualize laughter regulation as an expression-focused form of amusement regulation, relevant for understanding how people modulate positive emotional experiences in socially constrained contexts. Despite these challenges, little is known about how people successfully inhibit laughter in real time and how different regulatory strategies fare under varying social pressures.

While laughter enhances social bonding, relieves tension, and signals shared emotion, there are many situations in which its open expression is inappropriate or disruptive. From solemn public events to high-stakes professional settings, individuals often need to regulate laughter in order to align their emotional displays with contextual demands. Laughter regulation can therefore be viewed as a specific, expression-focused instance of amusement regulation—one that is particularly challenging due to the social contagion and involuntary nature of laughter[6]. Emotion regulation research has extensively examined how people manage negative emotional states such as anger and sadness[7,8]. Although the regulation of positive emotions such as happiness, pride, and hope has received increasing attention[9], the specific regulation of *amusement*—especially in socially dynamic contexts—remains comparatively underexplored. The present study addresses this gap by comparing amusement-elicited facial expressions with three common emotion-regulation strategies—cognitive reappraisal, expressive suppression, and distraction—during humorous experiences elicited by jokes. Experiments 1 and 2 tested these strategies in solitary listening contexts, examining their influence on both perceived joke funniness and facial muscle activity associated with smiling and laughter. Amusement regulation was operationalized through subjective funniness ratings and laughter-related EMG responses, which together index the expression and experience of amusement[10,11].

Cognitive reappraisal, expressive suppression, and distraction are among the most widely studied strategies for emotion regulation, particularly in the context of negative emotions[12–15]. Reappraisal involves reframing a situation to alter its emotional meaning; suppression refers to inhibiting outward emotional expressions; and distraction shifts attention away from the emotion-eliciting stimulus. While well characterized for downregulating negative emotions, these strategies have been studied far less in positive contexts. Findings on expressive suppression are especially inconsistent: some studies suggest it can dampen the subjective experience of positive emotions[16], while others find no such effect[17–19]. Examining how these strategies operate in the regulation of amusement therefore, extends this literature to a distinct, positive-valence domain where expressive and experiential components are tightly intertwined. Emotion regulation can target either the outward *expression* of emotion or the internal *experience* of emotion—often referred to as distinct regulatory goals[20,21]. These goals are not interchangeable: for example, expressive suppression aims to inhibit visible emotional displays without necessarily altering internal states, while distraction is more likely to dampen the emotional experience itself. Cognitive reappraisal may influence both. In the present study, we aimed to assess how these strategies affect both expression and experience by combining *fEMG recordings* as an index of amusement-related facial expression and *subjective funniness ratings* as an index of amusement experience. In addition, participants completed the PANAS scale before and after the experimental session to track potential overall mood shifts; these scores served as an exploratory check and were not analyzed by condition.

We selected these three regulation strategies because they are both theoretically distinct and empirically well-established. Expressive suppression aims to inhibit visible displays of amusement without necessarily changing one's internal state; cognitive reappraisal works by altering the emotional meaning of a stimulus, potentially modulating both experience and expression; and distraction disengages attention from the emotional source, often dampening internal experience more strongly than expressive behavior[13]. This range of mechanisms allowed us to probe how differently targeted strategies affect not only expressive outcomes (fEMG-based measures of laughter-related facial activity) but also evaluative outcomes—

specifically, how funny participants rated the same jokes under different regulation conditions. We used subjective funniness ratings as a proxy for the evaluative aspect of amusement, complementing fEMG-based measures of its expressive component. Together, these measures capture both the behavioral and cognitive–emotional outcomes of amusement regulation.

Our study specifically examined the down-regulation of amusement-related expressions and experiences, motivated by real-world situations where laughter is socially inappropriate or disruptive (e.g., professional or ceremonial contexts). While people also regulate amusement upward in affiliative or playful situations, our focus was on how individuals *suppress or attenuate* laughter-related expressions to meet situational demands. In this sense, laughter regulation serves as an expression-focused form of amusement regulation. To this end, we investigated how different strategies influence both facial expression and humor evaluation—first in solitary contexts (Experiments 1 and 2), and then under socially dynamic conditions (Experiment 3). Previous work has predominantly examined the suppression or reappraisal of humorous experiences in solitary settings[5,18,22], even though laughter is inherently social. It occurs more frequently in company than alone[3], serves as a social signal[23–25], and is highly contagious[6]. Observing others laugh can enhance the experience of amusement—a process known as social appraisal[5,26,27]—and makes suppression more difficult[28]. To capture this interpersonal dimension, we designed Experiment 3 to test how social feedback modulates amusement regulation through laughter. Specifically, we investigated whether hearing another person laugh makes it harder to suppress one's own facial responses and whether it changes the subjective experience of amusement. By introducing laughter feedback from others, Experiment 3 created a more ecologically valid and socially dynamic context for studying the regulation of amusement-related expressions.

To examine these processes objectively, we focused on the facial muscle activity that underlies laughter expression. Similar to smiling, laughter involves the coordinated activation of facial muscles, including the zygomatic major (lifting the corners of the mouth) and the orbicularis oculi (eye constriction), while the corrugator supercilii (brow furrowing) typically relaxes in response to positive stimuli[29,30]. Together, these muscle activations constitute the expressive component of amusement and laughter. To quantify amusement-related facial activity, we calculated a smile index combining activity from these muscles[31]. This composite measure served as our primary physiological index of amusement expression, used to assess the effects of different regulation strategies on spontaneous and laughter-related muscle activity across all three experiments.

The present study compared three laughter-regulation strategies—cognitive reappraisal, expressive suppression, and distraction—in their effects on facial expressions and amusement appraisal. In Experiments 1 and 2, participants listened to jokes in solitary listening contexts while applying these strategies, and we assessed both subjective humor ratings and laughter-related muscle activity. We hypothesized cognitive reappraisal to reduce both expression and amusement, expressive suppression to primarily reduce facial muscle activity, and distraction to dampen both expression and experience by diverting attention. In Experiment 3, we predicted that hearing another person's laughter would increase facial mimicry, undermine suppression success, and boost amusement ratings, highlighting the challenge of regulating amusement in socially dynamic contexts. While Experiments 1 and 2 examined regulation in solitary listening, Experiment 2 added a distraction condition, and Experiment 3 introduced social laughter feedback to create a more ecologically valid context. Together, these experiments provide a comprehensive test of how distinct emotion-regulation strategies modulate amusement-related expressions and experiences across solitary and social settings.

## Methods

We report how we determined our sample sizes, all data exclusions, all manipulations, and all relevant measures in the study. Pre-registration documents can be obtained under the following permanent identifiers. Experiment 1: https://doi.org/10.17605/OSF.IO/ZKHB8 (date: May 12,

2023); Experiment 2: https://doi.org/10.17605/OSF.IO/43V67 (date: April 2, 2024);

Experiment 3: https://doi.org/10.17605/OSF.IO/2SEUG (date: June 30, 2023).

## Method experiment 1

Experiment 1 focused on comparing the effects of cognitive reappraisal and expressive suppression on participants' evaluations of jokes and laughter-related muscle activity during solitary listening.

**Participants.** We recruited 44 participants from the immediate vicinity of the university in exchange for either 10€ or course credit. All participants were native German speakers with self-reported normal hearing. Four participants were excluded from the analysis due to technical problems during EMG recording, resulting in a final sample of $N = 40$ (30 females, 10 males; gender was obtained via participant self-report). Participants indicated their gender using the options *female*, *male*, *non-binary*, or *prefer not to say*. Information on participants' race or ethnicity was not collected, in line with institutional ethics guidance and given the homogeneity of the local student sample. Age ranged from 19 to 36 years ($M = 21.78$, $SD = 3.17$). All participants provided written informed consent, and the study was approved by the institutional ethics committee (2023-341). As preregistered, the target sample size ($N = 40$) was determined *a priori* based on Korb et al.[18], who examined how emotion-regulation strategies modulate smiling responses to affective stimuli using facial EMG. Their within-subject comparison of suppression and reappraisal yielded an estimated effect size of approximately $d \approx 0.6$, corresponding to a medium effect. Assuming comparable within-subject effects, a sample of 40 participants provides roughly 80% power to detect such differences. We therefore oversampled relative to the original $N = 21$ in Korb et al.[18] to ensure adequate power and replicability.

**Materials.** Auditory stimuli consisted of 100 jokes and 25 neutral facts, each presented in a question-answer format. Punchlines and answers were typically one or two words in length. Joke stimuli featured a humorous response, whereas neutral facts consisted of non-humorous but accurate information. All stimuli were recorded by four speakers (two female, two male), who were instructed to read the jokes in an amused and enthusiastic tone without laughing. Audio recordings are available upon request from the corresponding author. Stimulus examples and joke transcripts are available on the project's OSF repository (osf.io/y739x).

**Procedure.** Upon arrival at the laboratory, participants provided written informed consent and were briefed on the study procedure. Facial EMG electrodes were applied over the zygomaticus major (ZM), orbicularis oculi (OO), and corrugator supercilii (CS) muscles, following the guidelines by Fridlund and Cacioppo (1986). Participants then completed a baseline mood assessment using the German version of the Positive and Negative Affect Scale[32]; analysis concerning this measure is reported in the Supplement.

The experimental session began with an unconstrained baseline block consisting of 30 joke trials and 5 neutral fact trials. Each trial began with fixation cross (1000–2000 ms), followed by the first part of a joke or fact. A written reminder instructed participants that they could press the space bar whenever they wanted to hear the punchline, with a minimum delay of 1000 ms. The auditory punchline (variable duration) was then presented, followed by a blank screen appeared (3000 ms) and a funniness rating prompt ("How funny was the audio clip?") on a 5-point scale (1 = "not at all"; 5 = "very"). The order and assignment of audio files to conditions were randomized, and each clip was played only once per participant.

After a short break, participants completed two additional blocks with emotion-regulation instructions: one block with reappraisal and one with expressive suppression. The block order was fixed for all participants: the unconstrained baseline always preceded the regulation blocks, which were presented in the following sequence: expressive suppression, cognitive reappraisal (and, in Experiment 2, distraction). This ordering was chosen to minimize potential carryover effects between strategies. Specifically, starting with suppression—a surface-level, expression-focused strategy—reduced the risk that cognitively effortful strategies like reappraisal or distraction would influence responses in earlier blocks. It also ensured that participants first became familiar with the task in an unregulated, natural context before being asked to implement regulation strategies.

Each block followed the same trial structure as the unconstrained block but included condition-specific instructions (see Supplement). After each regulation block, participants rated the ease of strategy use on a 5-point scale (1 = very difficult, 5 = very easy). At the end of the experiment, participants completed a second PANAS and responded to open-ended questions about their strategy use.

**fEMG data preprocessing.** The EMG data were processed offline using *Brain Vision Analyzer 2.2* (Brain Products, Gilching, Germany). Raw signals were filtered with a 20 Hz low-cut and a 50 Hz notch filter to remove slow drifts and line noise, then full-wave rectified. Data segments were extracted from −500 ms to +4500 ms relative to the onset of the punchline. All recordings were screened for technical issues, including low activation due to loose or broken electrodes. Trials containing movement artifacts (e.g., coughing) were visually identified and excluded. For each trial, baseline correction was applied by subtracting the mean activity during the 500 ms pre-punchline baseline window. Averages per time bin were standardized within muscle (as recommended by[33] within each participant. Amusement-related facial activity was quantified using the *smile index*, calculated as the combined activity of the *orbicularis oculi* (OO) and *zygomaticus major* (ZM) minus *corrugator supercilii* (CS), following Mauersberger and Hess [31]: $[(OO + ZM) − CS]$.

**Statistical models.** All analyses were conducted on single-trial data using linear mixed-effects models (LMMs). Prior to model fitting, *smile index* values exceeding ± 3 *SD* from the participant's *M* were removed as outliers. Time was divided into 500-ms bins across the 4500 ms post-punchline window and treated as a continuous predictor. Data distributions were assumed to be approximately normal; visual inspection of residuals indicated no major violations of normality or homogeneity of variance. Detailed assumption checks are available in the open online materials for each model.

To statistically test all predicted effects, linear mixed models with maximal random-effects structures were employed, including all theoretically justified random slopes[34,35], using the function 'lmer' of the package lme4 (version 1.1-29). To assess the contribution of the fixed effects of interests, we compared full models to null models excluding predictors of interest but retaining the same random structure[36]. The relative contribution of each predictor was additionally tested using likelihood-ratio tests[37], by comparing all possible models and sequentially dropping single terms using the function 'mixed' of the package afex (version 1.1-1).

The total variance explained by the fixed and random effects were obtained using the marginal and conditional $R^2$ respectively[38] using function 'r.squaredGLMM' of the package MuMIn (version 1.47.5). Using the same method, the unique contribution of individual predictors was assessed by calculating partial $R^2$ along with their confidence intervals using the 'r2beta' function of the package r2glmm (version 0.1.2). Finally, the significance of individual fixed effects was evaluated using the Satterthwaite approximation[39] implemented in the package lmerTest (version 3.1-3), with models fitted using restricted maximum likelihood. All *p*-values for post hoc contrasts between conditions were Bonferroni-corrected.

## Method experiment 2

Experiment 2 aimed to replicate the findings of Experiment 1 while extending the design by including a third regulation strategy—distraction—alongside cognitive reappraisal and expressive suppression, again during solitary joke listening.

**Participants**. Forty-two participants were recruited from the local university community in exchange for either 10€ or course credits. All participants were native German speakers with self-reported normal hearing. Two participants were excluded due to technical problems during EMG data collection, resulting in a final sample of $N = 40$ (33 female, 7 male; gender obtained via self-report). Participants indicated their gender using the options *female, male, non-binary,* or *prefer not to say*. Information on participants' race or ethnicity was not collected, in line with institutional ethics guidance and given the homogeneity of the local student sample. Age ranged from 19 to 27 years ($M = 21.75$, $SD = 2.22$). All participants provided written informed consent, and the study was approved by the institutional ethics board of the University of Göttingen (approval ID: 2023-341). The target sample size ($N = 40$) and power rationale followed the same logic as described for Experiment 1 (see above), based on the expected medium within-subject effect size ($d \approx 0.6$; Korb et al., 2012). The final sample size was therefore identical to that used in Experiment 1 for direct comparability.

**Materials**. The auditory jokes were identical to those used in Experiment 1. For the distraction condition, a visual search task was introduced using a hidden-object poster positioned ~60 cm in front of the participant on a room divider. Participants were instructed to engage with the poster while listening to the jokes (see Supplement for full instruction text). The poster image is not reproduced here due to copyright.

**Procedure**. The overall procedure followed that of Experiment 1, with a few modifications. Each experimental block consisted of 20 joke trials, and neutral facts were omitted. Participants completed four blocks in total: an initial unconstrained baseline block, followed by three laughter-regulation blocks with instructions for cognitive reappraisal, expressive suppression, or distraction (see Supplement). The distraction block involved performing the visual hidden-object task described in the Materials section. The block order for the three regulation conditions was fixed across participants.

**fEMG data preprocessing and statistical models**. The fEMG preprocessing and statistical analysis procedures were identical to those described for Experiment 1. For details on signal filtering, baseline correction, calculation of the *smile index*, and the linear mixed-effects modeling approach, see the corresponding sections above.

**Method experiment 3**
Experiment 3 extended the previous designs by introducing a more ecologically valid, socially dynamic context. Participants listened to jokes accompanied by prerecorded laughter from others, allowing us to examine how social laughter feedback influences amusement-related facial activity and its regulation. This experiment compared expressive suppression with an unconstrained baseline condition, focusing on the regulation of amusement in response to social laughter cues. The focus on suppression followed the rationale outlined in the Introduction, as this strategy can be implemented rapidly in interactive contexts.

**Participants**. Forty-one participants (29 female, 11 male, 1 non-binary; age range = 19–35, $M = 23.4$, $SD = 3.51$) were recruited from the University of Göttingen in exchange for 8 € or course credits. All were native German speakers with self-reported normal hearing. Gender was obtained via self-report using the same options as in Experiments 1 and 2 (*female, male, non-binary, prefer not to say*); information on race or ethnicity was not collected, in line with institutional ethics guidance. All participants provided written informed consent and were debriefed after the experiment. The study was approved by the institutional ethics committee of the University of Göttingen (approval ID: 2023-344).

In the absence of prior studies directly testing this paradigm, the target sample size was matched to Experiments 1 and 2 to ensure comparable statistical power and facilitate cross-experiment analyses.

**Materials**. The auditory joke and fact stimuli were identical to those used in Experiments 1 and 2; neutral facts were not included in this experiment. Social feedback was presented via short video clips depicting a male or female model displaying either laughter or a neutral facial expression. In total, 14 laughing clips of a female model and 11 of a male model were used, along with 10 neutral-expression clips from each model. All videos captured the models' face and shoulders from a frontal perspective and were edited to a uniform duration of 3 s. Audio levels were normalized for consistent loudness within each stimulus type.

After the main experiment, participants rated the models on likability ($M = 4.09$, $SD = 0.80$), authenticity of laughter ($M = 3.73$, $SD = 0.91$) on 5-point Likert scales (1 = "not at all", 5 = "very"). Example video files are available from the corresponding author upon request.

**Procedure**. Participants completed one regulation condition (expressive suppression) and one baseline condition (unconstrained), each combined with either laughter or neutral social feedback, resulting in four combinations. Each combination was presented 15 times, yielding 60 joke trials. Ten fact trials were included in suppression blocks only.

The trial structure was identical to that in Experiments 1 and 2, except that a 3 s video clip (laughing or neutral) was presented after the punchline segment, followed by a 3000-ms blank screen and a 500 ms fixation cross. Video and audio file assignments were counterbalanced across participants.

After the main experiment, participants rated the likeability of the individuals shown in the feedback videos and the authenticity of their laughter. They then completed a post-test PANAS assessment and responded to two exploratory questions: (1) "How difficult was it for you to inhibit your laughter during this study?", and (2) "How difficult is it for you to inhibit your emotions in everyday life situations?" (1 =" very difficult", 5 = "very easy").

**fEMG data preprocessing and statistical models**. fEMG preprocessing procedures paralleled those described for Experiment 1 and 2. Data were segmented from 1000 ms before the punchline onset (baseline) to 6000 ms after the onset of the social feedback video. Muscle onsets were identified as activations in the *zygomaticus major* (ZM) exceeding 3 $SDs$ above baseline, with a minimum peak of 30 μV. Statistical analyses followed the same linear mixed-effects modeling approach as in Experiments 1 and 2. LMMs included maximal random-effects structures where possible; in cases of convergence issues, random structures were simplified as needed.

## Results
### Changes in the subjective experience of amusement during regulation (H1) indexed via *ratings of funniness (Exp. 1)*
We analyzed subjective funniness ratings using a linear mixed-effects model (LMM) with *condition* (unconstrained/suppression/reappraisal) as fixed effect and random intercepts for *joke ID* and *participant ID*. Jokes in the reappraisal condition received significantly lower ratings ($M = 2.08$, $SD = 1.20$) than those in the unconstrained condition ($M = 2.27$, $SD = 1.17$), $\beta = -0.21$, $CI_{boot} = [-0.39; -0.04]$, $SE = 0.09$, $t = -2.30$, $p = .027$. Ratings in the suppression condition ($M = 2.25$, $SD = 1.24$) did not differ significantly from the unconstrained condition, $\beta = -0.04$, $CI_{boot} = [-0.16; 0.08]$, $SE = 0.06$, $t = -0.57$, $p = 0.574$, $p_{adj} = 1.000$, whereas reappraisal and suppression differed marginally (contrast (reappraisal—suppression): $\beta = -0.18$, $CI_{asymp} = [-0.38; 0.02]$, $SE = 0.08$, $z = -2.13$, $p_{adj} = 0.0995$).

### Changes in the subjective experience of amusement during regulation (H1) indexed via *ratings of funniness (Exp. 2)*
We analyzed subjective funniness ratings using a LMM with *condition* (unconstrained/suppression/reappraisal/distraction) as fixed effect and random intercepts for *participant ID* and *joke ID*. Compared to the unconstrained condition ($M = 2.01$, $SD = 1.04$), both reappraisal ($M = 1.70$, $SD = 0.96$; $\beta = -0.30$, $CI_{boot} = [-0.46; -0.15]$, $SE = 0.08$, $t = -3.74$, $p = 0.001$) and distraction ($M = 1.60$, $SD = 0.84$; $\beta = -0.40$, $CI_{boot} = [-0.53$;

−0.28], $SE = 0.06$, $t = −6.27$, $p < 0.001$) significantly reduced humor ratings. Suppression led to a small, non-significant reduction compared to the unconstrained condition ($M = 1.89$, $SD = 1.02$; $β = −0.11$, $CI_{boot} = [−0.23; 0.02]$, $SE = 0.06$, $t = −1.84$, $p = .071$).

A significant difference between regulation strategies was only apparent for distraction and suppression (contrast $_{(distraction—suppression)}$: $β = −0.29$, $CI_{asymp} = [−0.44; −0.14]$, $SE = 0.06$, $z = −5.05$, $p_{\_adj} <.001$), but not between distraction and reappraisal (contrast $_{(distraction—reappraisal)}$: $β = −0.11$, $CI_{asymp} = [−0.28; 0.07]$, $SE = 0.07$, $z = −1.57$, $p_{\_adj} = 0.692$) or reappraisal and suppression (contrast $_{(reappraisal—suppression)}$: $β = -0.18$, $CI_{asymp} = [−0.39; 0.03]$, $SE = 0.08$, $z = −2.28$, $p_{\_adj} = 0.134$).

### Changes in muscle activity during regulation (H2) indexed via fEMG activity over time (Exp. 1)

To assess muscle activity changes during joke perception over time, we analyzed trial-based averages of the smile index across time bins using a LMM. Fixed effects included *condition* (unconstrained/suppression/reappraisal) and *time* (z-standardized), with random intercepts for *joke ID* and *participant ID*. Compared to the unconstrained condition, smile-related muscle activity over time was significantly reduced in the reappraisal condition ($β = −0.11$, $CI_{boot} = [−0.15; −0.07]$, $SE = 0.02$, $t = −5.35$, $p < .001$) and even more for the suppression condition ($β = −0.17$, $CI_{boot} = [−0.21; −0.13]$, $SE = 0.02$, $t = −8.48$, $p < 0.001$; see Fig. 1A). Further, post-hoc comparison revealed reduced smile index over time for suppression in comparison to the reappraisal condition (contrast $_{(reappraisal—suppression)}$: $β = 0.64$, $CI_{asymp} = [0.02; 0.11]$, $SE = 0.02$, $z = 3.69$, $p_{adj} < 0.001$).

To investigate whether reappraisal increased negative emotional response, we analyzed CS muscle activity using a separate LMM (H2b). Contrary to our prediction, CS activity was highest during the unconstrained condition. Compared to unconstrained listening, CS activation was significantly lower during reappraisal ($β = −0.05$, $CI_{boot} = [−0.09; −0.01]$, $SE = 0.02$, $t = −2.12$, $p = 0.041$) and did not significantly differ during suppression ($β = −0.02$, $CI_{boot} = [−0.06; 0.02]$, $SE = 0.02$, $t = −1.20$, $p = 0.241$).

### Changes in muscle activity during regulation (H2) indexed via fEMG activity over time (Exp. 2)

Smile-related muscle activity over time was analyzed using a linear mixed-effects model with *condition* (unconstrained, suppression, reappraisal, distraction) and *time* (z-standardized) as fixed effects, and random intercepts for *participant ID* and *joke ID*. Average muscle activity over time was highest in the unconstrained condition, and significantly reduced in all regulation conditions: Reappraisal, $β = -0.34$, $CI_{boot} = [-0.48; -0.21]$, $SE = 0.07$, $t = −4.90$, $p < 0.001$, and distraction, $β = -0.36$, $CI_{boot} = [−0.52; −0.23]$, $SE = 0.07$, $t = −4.99$, $p < 0.001$, showed comparable reductions, while suppression produced the largest decrease, $β = −0.45$, $CI_{boot} = [−0.58; −0.33]$, $SE = 0.06$, $t = −7.30$, $p < 0.001$. See Fig. 1C for the time course of muscle activity by condition. Further, post-hoc comparisons did not reveal any significant differences for smile index over time between suppression, reappraisal and distraction (contrast $_{(reappraisal—suppression)}$: $β = 0.11$, $CI_{asymp} = [−0.02; 0.24]$, $SE = 0.05$, $z = 2.29$, $p_{adj} = .133$; contrast $_{(distraction—suppression)}$: $β = 0.08$, $CI_{asymp} = [−0.03; 0.20]$, $SE = 0.04$, $z = 1.92$, $p_{adj} = 0.328$; contrast $_{(distraction—reappraisal)}$: $β = −0.03$, $CI_{asymp} = [−0.18; 0.12]$, $SE = 0.06$, $z = −0.46$, $p_{adj} = 1.000$).

To investigate whether reappraisal increased negative emotional response, we analyzed CS muscle activity using a separate LMM (H2b). Compared to unconstrained listening none of the condition effects reached significance, reappraisal ($β = −0.04$, $CI_{boot} = [−0.10; 0.02]$, $SE = 0.03$, $t = −1.15$, $p = 0.259$), CS suppression ($β = −0.01$, $CI_{boot} = [−0.10; 0.02]$, $SE = 0.03$, $t = −0.42$, $p = 0.682$), distraction ($β = −0.08$, $CI_{boot} = [−0.15; 0.01]$, $SE = 0.04$, $t = −1.91$, $p = 0.065$), over time.

### Relation between experienced amusement and fEMG activity (Exp.1).

To assess the relationship between perceived funniness and facial responses during unconstrained listening, we analyzed the trial-based smile index (z-standardized) as a function of *time* (z-standardized) and *funniness ratings* (on a 1-5 scale) using a LMM with random intercepts of *joke ID* and *participant ID*. Higher funniness ratings were associated with increased smile-related muscle activity over time. For example, compared to jokes rated 1, jokes rated 2 showed a moderate increase in smile-related fEMG activity ($β_{2-1} = 0.27$, $CI_{boot} = [0.15; 0.40]$, $SE = 0.07$, $t = 4.16$, $p <0.001$); while jokes rated 5 showed a much steeper increase ($β_{5-1} = 1.47$, $CI_{boot} = [1.08; 1.84]$, $SE = 0.16$, $t = 9.09$, $p <0.001$). Full parameter estimates are reported in Supplement.

Building on this finding, we next examined whether the relationship between smile-related muscle activity and perceived funniness differed across laughter regulation strategies. We conducted a LMM with funniness ratings as the dependent variable, and *smile index* (z-standardized), *condition* (unconstrained/ suppression/ reappraisal), and their interaction as fixed effects. Random intercepts for *joke ID* and *participant ID* were included. The association between smiling and funniness ratings was strongest in the suppression condition ($β = 0.12$, $CI_{boot} = [0.03; 0.20]$, $SE = 0.04$, $t = 2.75$, $p = 0.009$), suggesting that even small facial reactions during suppression were linked to higher subjective amusement. No significant differences in this relationship were observed between the reappraisal and unconstrained conditions ($β = 0.02$, $CI_{boot} = [−0.03; 0.08]$, $SE = 0.03$, $t = 0.91$, $p = 0.371$). These condition-specific associations are illustrated in Fig. 1B.

### Relation between experienced amusement and fEMG activity (Exp. 2).

To assess whether funniness ratings predicted facial responses in the unconstrained condition, we analyzed the trial-based smile index (z-standardized) as a function of *time* (z-standardized) and funniness ratings (1–5 scale) using a linear mixed-effects model with random intercepts for participant and joke ID. Higher funniness ratings were associated with increased smile-related muscle activity over time. For example, jokes rated 2 showed a modest increase compared to jokes rated 1 ($β_{2-1} = 0.32$, $CI_{boot} = [0.14; 0.52]$, $SE = 0.10$, $t = 3.34$, $p = 0.002$), while jokes rated 5 produced a substantially larger increase ($β_{5-1} = 2.56$, $CI_{boot} = [1.61; 3.64]$, $SE = 0.38$, $t = 6.78$, $p <0.001$). Full parameter estimates are reported in Supplement.

We next examined whether the relationship between smile-related muscle activity and funniness ratings varied across regulation strategies. Using a LMM, we entered funniness ratings as the dependent variable, with *smile index* (z-standardized), *condition* (unconstrained, suppression, reappraisal, distraction), and their interaction as fixed effects. Random intercepts for *participant ID* and *joke ID* were included. The effect of muscle activity on funniness ratings was strongest in the suppression condition ($β = 0.15$, $CI_{boot} = [0.05; 0.27]$, $SE = 0.06$, $t = 2.77$, $p = 0.009$), indicating that even minimal smiling was associated with increased amusement when participants attempted to suppress their reactions. No significant differences were found between the unconstrained and reappraisal ($β = 0.00$, $CI_{boot} = [−0.06; 0.06]$, $SE = 0.03$, $t = 0.11$, $p = 0.910$) or distraction conditions ($β = −0.03$, $CI_{boot} = [−0.08; 0.02]$, $SE = 0.03$, $t = −1.22$, $p = 0.230$), respectively. Fig. 1D shows these condition-specific relationships.

### Influence of social feedback on amusement (results experiment 3)

**Changes in the subjective experience of amusement due to social feedback (H1) indexed via *ratings of funniness (Exp. 3)*.** We analyzed participants' funniness ratings as the dependent variable in a LMM with *response condition* (unconstrained/ suppression) and *feedback type* (laughter/neutral) as fixed effects, and random intercepts for *participant ID* and *joke ID*. Jokes followed by laughter feedback were rated as significantly funnier than those followed by neutral feedback ($β = 0.31$, $CI_{boot} = [0.19; 0.42]$, $SE = 0.06$, $t = 4.88$, $p < 0.001$). Jokes in the unconstrained condition were also rated higher than those in the suppression condition ($β = 0.14$, $CI_{boot} = [−0.01; 0.26]$, $SE = 0.07$, $t = 2.04$, $p = 0.045$). The interaction between feedback type and response condition was not

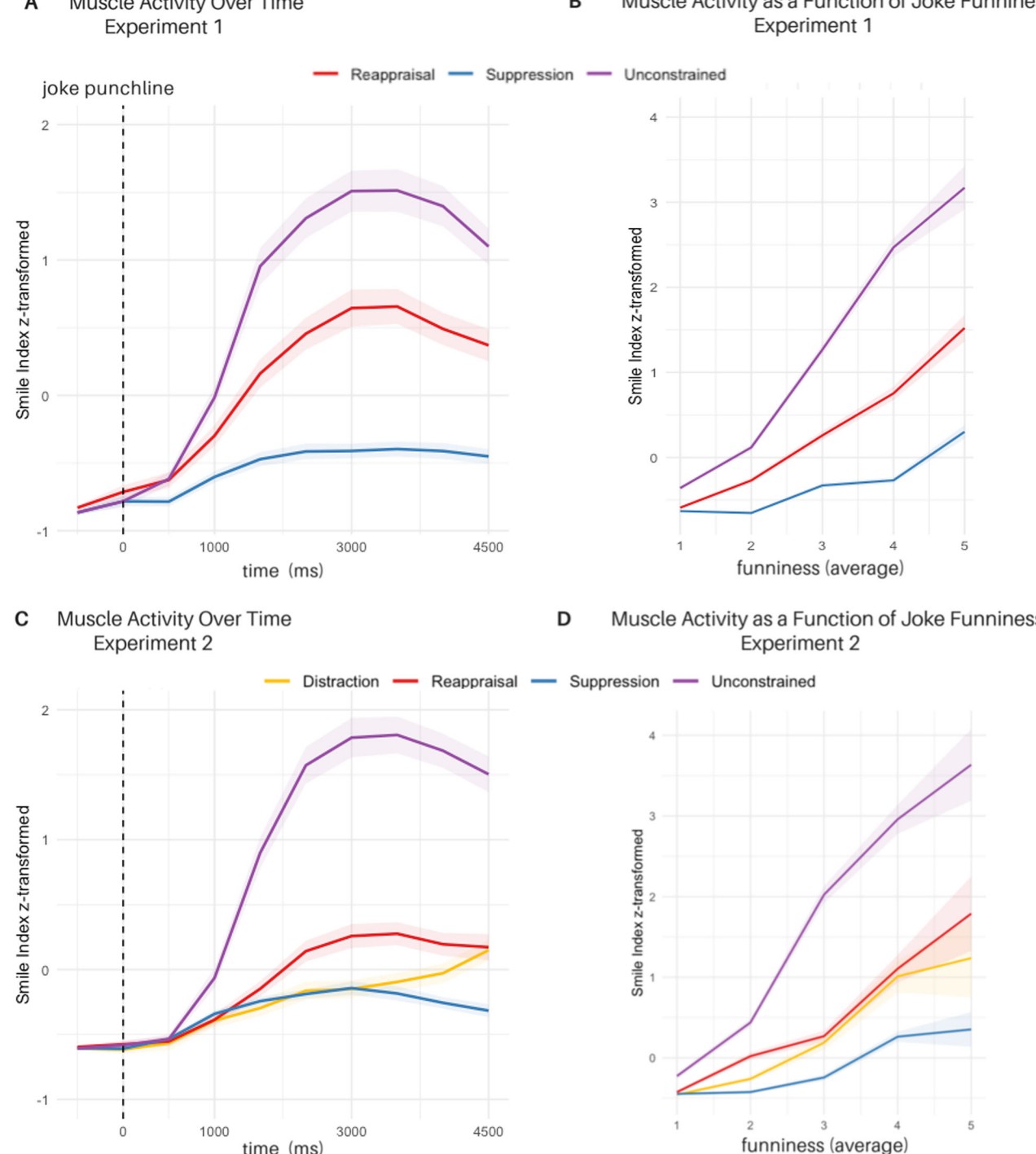

**Fig. 1 | Overview of EMG activity across Experiments 1 and 2. A**, **B** display results from Experiment 1 ($N = 44$). **A** shows average smile-related muscle activity over time (0–4500 ms after punchline onset) by condition (unconstrained, reappraisal, suppression). Panel **B** shows mean smile index values across the full time window, plotted by condition and funniness rating (1 = least funny, 5 = most funny). **C**, **D** display results from Experiment 2 ($N = 42$). **C** shows average smile-related muscle activity over time by condition (unconstrained, reappraisal, suppression, distraction). **D** shows mean smile index values by condition and funniness rating. Shaded areas represent standard errors. The smile index reflects the combined activity of the orbicularis oculi (OO) and zygomaticus major (ZM), minus corrugator supercilii (CS): [(OO + ZM) − CS].

significant ($\beta = 0.03$, $CI_{boot} = [-0.11; 0.17]$, $SE = 0.07$, $t = 0.41$, $p = 0.684$). Descriptive values are presented in Table 1; Fig. 2A provides a visual summary.

**Influence of laughter feedback on inhibition effectiveness (H2; Exp. 3).** In suppression trials, we evaluated facial muscle responses using two approaches: (1) onset ratios, defined as the proportion of trials with zygomaticus major (ZM) activations exceeding 3 SDs above baseline (≥30 µV), and (2) baseline-corrected smile index over time. Laughter feedback led to significantly more smile onsets ($M = 0.14$, $SD = 0.19$) than neutral feedback ($M = 0.08$, $SD = 0.10$; $t(40) = 2.56$, $p = 0.014$, $d = 0.40$, $CI = [0.08; 0.72]$); Fig. 2B provides a visual summary.

To analyze smile index over time, we used a linear mixed-effects model with feedback type (laughter, neutral), time (1 s bins across 6 s; z-standardized), and their interaction as fixed effects, and a random intercept for participant with random slopes for feedback (all within the suppression condition). Muscle activity increased significantly over time in the laughter feedback condition ($\beta = 0.09$, $CI_{boot} = [0.03;0.16]$, $SE = 0.03$, $p = 0.011$) compared to the neutral condition. See Fig. 3B for time courses.

**Influence of humorous stimuli on mimicry suppression (H3; Exp. 3).** To examine how stimulus content affected smile-related mimicry during suppression, we analyzed smile-index trajectories following jokes versus neutral facts. Baseline correction was shifted to 1 second prior the punchline onset to capture stimulus-elicited activity. A LMM included *stimulus type* (joke/fact), *time* (9 time bins over 6 s, z-standardized), and their interaction as fixed effects, with random intercept for *participant ID* and a random slope of *stimulus type*. The estimated smile index was significantly higher following joke stimuli than fact stimuli ($\beta = 0.55$, $CI_{boot} = [0.22;0.84]$, $SE = 0.16$, $p = 0.002$). However, there was no significant interaction between *stimulus type* and *time* ($\beta = 0.05$, $CI_{boot} = [-0.19;0.08]$, $SE = 0.07$, $p = 0.419$), indicating that smiling increased over time regardless of stimulus type ($\beta = 0.21$, $CI_{boot} = [0.10;0.34]$, $SE = 0.06$, $p = 0.001$).

## Discussion

Expressions of amusement, such as laughter and smiling, play an important role in human social relationships, serving as nonverbal signals of positive emotion, affiliation, and approachability. These expressions foster social bonds and defuse tension. However, in settings where laughter is inappropriate, it can lead to misunderstandings and discomfort, potentially damaging relationships, undermining the seriousness of the situation, or appearing disrespectful. The ability to control or suppress laughter is therefore an important skill in socially sensitive contexts.

Across three experiments, we investigated how different strategies affect laughter regulation and humor evaluations. In Experiments 1 and 2, participants listened to jokes while applying cognitive reappraisal, expressive suppression, or distraction, or responded naturally without constraints. As predicted, distraction and suppression were most effective at reducing laughter-related facial muscle activity, while reappraisal produced smaller but reliable reductions. In line with our expectations, reappraisal—but not suppression—consistently decreased participants' funniness ratings, indicating its selective impact on the cognitive evaluation of humor. Also consistent with our hypotheses, suppression appeared more effective for mildly humorous stimuli but became less successful as perceived funniness increased. Experiment 3 introduced social laughter feedback to examine how the presence of another's laughter modulates mimicry and regulation.

While suppression and distraction proved effective in solitary contexts (Experiments 1 and 2), Experiment 3 showed that the presence of another person's laughter significantly impaired participants' ability to suppress their own facial responses. This was reflected in both a higher frequency of muscle activation onsets and greater overall laughter-related activity. These results align with prior work demonstrating that smile mimicry is difficult to inhibit when social cues are present[28], and support suggestions that automatic mimicry can be modulated—but not fully overridden—by top-down regulatory control[40,41].

A pooled analysis of Experiments 1 and 2 indicated that expressive suppression had no significant impact on participants' funniness ratings, even with increased statistical power (see Supplement for pooled analysis)—consistent with previous findings[17–19]. In contrast, Experiment 3 revealed that suppression in the presence of laughter feedback led to a reduction in perceived funniness. This suggests that the presence of others not only hampers amusement-related expressions but also alters how humorous content is evaluated.

Different mechanisms may underlie this shift in experience. Suppressing visible emotional reactions in social settings may increase internal stress or discomfort[42], though whether this generalizes to prerecorded feedback remains unclear. Alternatively, the need to inhibit mimicry in real time may create cognitive conflict and effort, both of which are associated with reductions in positive emotional experience[43–45]. Moreover, the trial-wise structure of Experiment 3—with regulation instructions varying across trials—may have introduced an additional regulatory load compared to the block-wise structure in Experiments 1 and 2. Future research should explore how regulatory dynamics, effort, and context-specific stress influence emotional responses in social settings.

Taken together, our findings highlight the complexity of emotion regulation in social contexts, especially when automatic mimicry tendencies clash with intentional control goals. While distraction and

### Table 1 | Descriptive Values of Funniness Ratings in Experiment 3 (Jokes only)

| Feedback type | Response Condition | Mean | SE | 95% Confidence Interval | |
|---|---|---|---|---|---|
| | | | | Lower | Upper |
| Laughter | Suppression | 2.25 | 0.095 | 2.06 | 2.44 |
| Neutral | Suppression | 1.93 | 0.095 | 1.74 | 2.13 |
| Laughter | Unconstrained | 2.41 | 0.095 | 2.22 | 2.60 |
| Neutral | Unconstrained | 2.07 | 0.095 | 1.88 | 2.26 |

**A Funniness Self-Report**

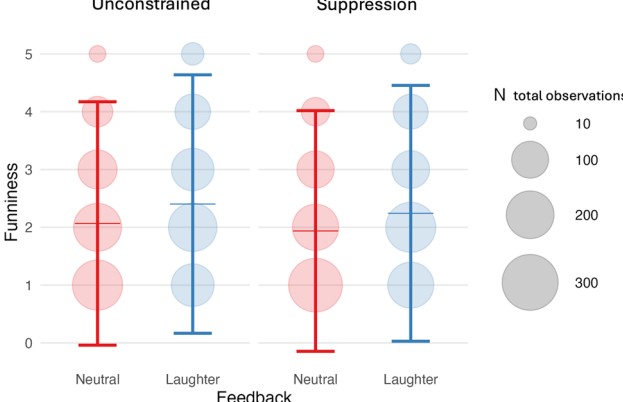

**B Muscle Onset Ratios**

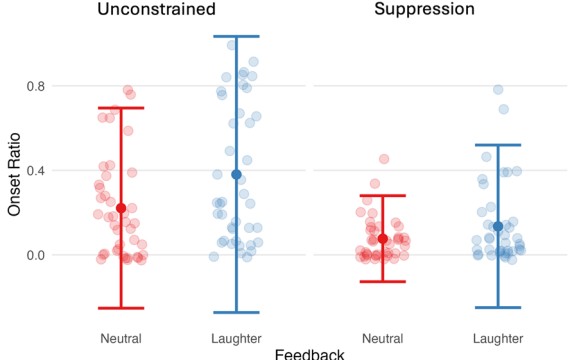

**Fig. 2 | Overview of funniness and onset ratios in experiment 3. A**: mean ratings of funniness divided by condition. **B**: The onset ratios reflect the sum of ZM muscle onsets above a 3 *SD* threshold from baseline with a minimum of 30μV peak activation, divided by the sum of all trials. An onset ratio of 1 would therefore reflect laughing related ZM activity in all trials. Error bars represent *SD*. *N* corresponds to the combined number of ratings across all participants. *N* = 41 participants.

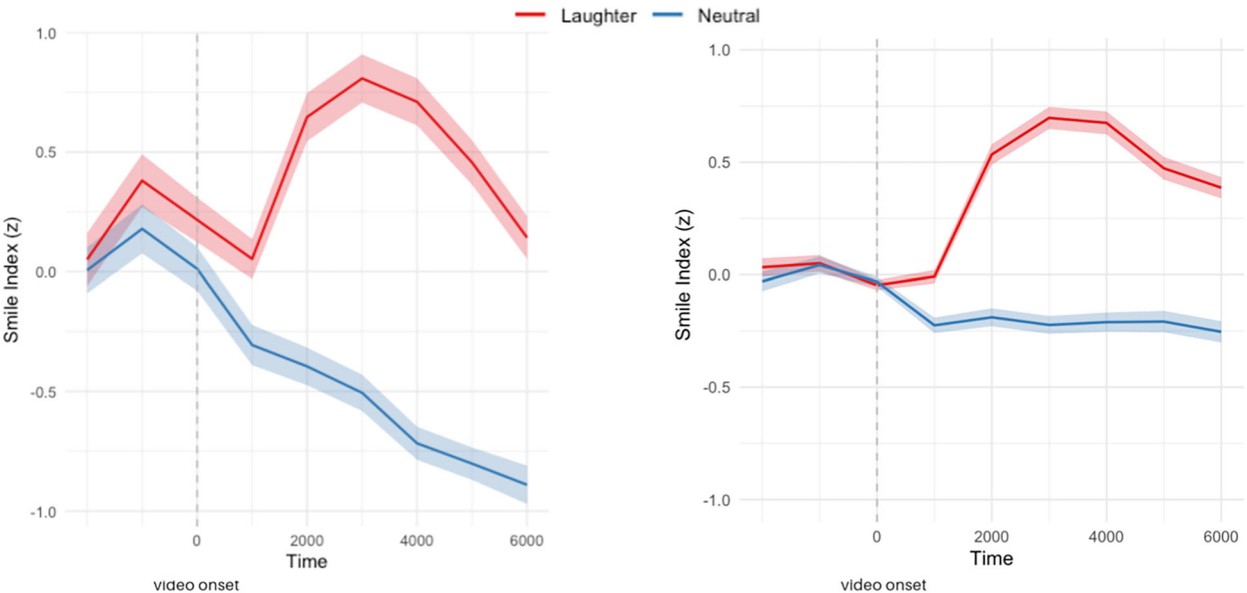

**A** Muscle Activity Over Time during **Unconstrained** Trials

**B** Muscle Activity Over Time during **Suppression** Trials

**Fig. 3 | Time course of smile-related muscle activity from Experiment 3. A**, **B** show mean smile index values (without baseline correction) during suppression trials, aligned to the onset of the social feedback video (dashed line), and separated by condition and feedback type (laughter vs. neutral). This visualization includes muscle responses to both the joke and the subsequent feedback. Shading corresponds to *SE*. Smile indices reflect the average muscular pattern of smiling $[(OO + ZM)\text{-}CS]$. $N = 41$ participants.

suppression can attenuate facial expressions of amusement in solitary settings, the presence of another person's laughter fundamentally alters the regulatory demands. This underscores the social embeddedness of emotion regulation: strategies that are effective in isolation may lose their potency or produce different outcomes when applied in interactive environments. These findings have broader implications for understanding emotional self-regulation in everyday social life. Situations that demand seriousness—such as professional, medical, or ceremonial contexts—often occur in the presence of others, where social cues can both reinforce and challenge internal control. Our results suggest that attempts to inhibit laughter in such settings may be more cognitively taxing and emotionally consequential than previously assumed. This resonates with the view that social emotion regulation involves a balancing act between personal goals, interpersonal norms, and the automatic pull of shared emotional states.

Although theoretical models increasingly emphasize the dynamic nature of emotion regulation, both within and between individuals[46,47], empirical studies that directly examine regulatory processes in social interactions remain scarce. Our findings contribute to filling this gap by demonstrating how social context shapes both emotional expression and subjective experience during attempts to regulate amusement. In particular, the presence of another person's laughter intensified emotional contagion and undermined regulatory success, even when participants were explicitly instructed to suppress their reactions.

Our findings also nuance the relationship between emotional expression and experience, contributing to debates surrounding the facial feedback hypothesis[48]. While this theory proposes that facial expressions can influence felt amusement, empirical support has been mixed[49]. In our study, suppression of facial responses did not reduce amusement ratings in solitary contexts—suggesting limited feedback effects under these conditions. However, in social contexts, suppression coincided with reduced funniness ratings, pointing to a more complex interplay between expression, experience, and social cues. Rather than a simple causal link from expression to emotion, these results support the idea that the emotional impact of facial feedback may depend on broader situational factors, including whether emotions are shared or socially reinforced.

## Limitations

While the present study provides insights into amusement regulation, several avenues remain open for future research. Although we used standardized jokes and controlled feedback, the artificial setting may not fully capture the dynamics of live social interaction. One consideration is that the mean funniness ratings observed in our experiments were moderate in absolute terms. However, such values are common in laboratory-based humor research using ethically appropriate and experimentally constrained materials[10,18,50]. This convergence suggests that our ratings reflect the typical range of amusement elicited in controlled settings, rather than a lack of stimulus validity. In addition, regulation blocks followed a fixed sequence to reduce strategy contamination, but future studies should test whether counterbalancing affects regulation success or fatigue; our exploratory analyses, including trial number as a predictor did not reveal patterns consistent with a simple fatigue or habituation explanation. Participants were aware of regulation instructions, which may have shaped their engagement with the stimuli. In addition, while we monitored general mood changes across the session using pre- and post-experimental PANAS ratings, the design did not permit condition-specific emotion assessments. Future work could implement more fine-grained, repeated mood measures (e.g., brief probes after each block) to complement trial-level funniness ratings and track broader emotional shifts over time. Moreover, the identity and familiarity of the laughter source were not varied; exploring how perceived social closeness influences regulatory outcomes would enrich ecological validity. Importantly, we focused exclusively on the down-regulation of amusement—reflecting contexts in which laughter may be inappropriate or disruptive. Yet people also up-regulate laughter to foster affiliation and shared emotion. Investigating the use and effectiveness of amusement up-regulation strategies under varying social conditions represents a promising direction for future research. In addition, as Experiment 3 included only expressive suppression, future research should investigate whether distraction or reappraisal strategies can be adapted to similarly dynamic social contexts, perhaps through preparatory cues or longer response windows. Finally, interindividual factors—such as habitual strategy use, social anxiety, or sensitivity to social norms—may shape regulation success. Understanding these influences, along with how suppressed laughter is socially

**Article**

interpreted (e.g., as disapproval or restraint), could clarify the interpersonal consequences of emotional inhibition in everyday settings. Beyond these conceptual considerations, a closer look at the physiological data offers additional insight into the underlying expressive dynamics. A visual inspection of the muscle activity time course suggests that highly amusing stimuli elicited activation not only in ZM and OO (associated with smiling) but also in CS. Contrary to expectations, CS activity did not increase during reappraisal, possibly because the dominant smile response overshadowed more subtle brow-related effort signals. This pattern aligns with prior evidence that intense smiles can involve multiple facial regions[51] and that CS activation may reflect both cognitive effort and the intensity of expression[52].

## Conclusions

Together, these findings underscore the social nature of emotion regulation and highlight the particular challenges of suppressing shared positive amusement expressed through laughter. They show that the effectiveness and consequences of regulation strategies depend not only on the type of emotion but also on the social context in which regulation takes place. Understanding how we navigate such moments of emotional conflict offers valuable insight into the broader mechanisms that support social functioning and self-control, particularly in situations demanding emotional restraint.

## Data availability

Data files used in the reported analysis are available under osf.io/prhyx/ (Exp. 1 + 2) and osf.io/u58by (Exp3). All auditory materials as well as raw physiological data can be made available upon request.

## Code availability

All analysis scripts are available under osf.io/prhyx/ (Exp. 1 + 2) and osf.io/u58by (Exp3). The full experimental code can be made available upon request.

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

## Acknowledgements

The authors thank Luisa Engelke, Florian Burkhardt and Kerstin Braun for their help with data collection and audio/video cutting, Matilda Rabenstein and Ralph Hertwig for fruitful discussions, and Anand Krishna for directing the voice actors and recording the audio materials. This research was supported by a Leibniz ScienceCampus Primate Cognition grant and by the Deutsche Forschungsgemeinschaft (DFG, German Research Foundation) - Project-ID 454648639 - SFB 1528. The funders had no role in study design, data collection and analysis, decision to publish or preparation of the manuscript.

## Author contributions

V.M. conceptualized the study, curated the data, conducted formal analyses, developed the methodology, created visualizations, and drafted the manuscript. A.Z. conducted formal analyses and contributed to methodology, visualization, and writing the manuscript. S.M. conducted formal analyses and produced visualizations. A.S. conceptualized the study, developed the methodology, supervised the project and contributed to the writing process. All authors approved the final manuscript.

## Funding

## Competing interests

The authors declare no competing interests.
