## [Transparent Peer Review file · Communications Psychology]

Laughter regulation in solitary and social contexts varies across emotion regulation strategies.

Corresponding Author: Dr Vanessa Mitschke

Version 0:

Decision Letter:

Dear Dr Mitschke,

Thank you for your patience during the peer-review process. Your manuscript titled "Laughter Regulation in Solitary and Social Contexts: Differential Effects of Suppression, Reappraisal, and Distraction" has now been seen by 2 reviewers, whose comments are appended below. You will see that they find your work of some potential interest. However, they have raised quite substantial concerns that must be addressed. In light of these comments, we cannot accept the manuscript for publication, but would be interested in considering a revised version that fully addresses these serious concerns.

We hope you will find the Reviewers' comments useful as you decide how to proceed. Should additional work allow you to address these criticisms, we would be happy to look at a substantially revised manuscript. If you choose to take up this option, please highlight all changes in the manuscript text file, and provide a detailed point-by-point reply to the reviewers.

Editorially, we consider the following issues critical to a successful revision:

- 1) The reviewers' methodological concerns must be satisfactorily addressed; responding to their concerns about the PANAS analysis and counterbalancing/block order concerns may potentially require an additional experiment.
- 2) Multiple comparisons yield non-significant results, which are interpreted as an absence of a difference between conditions. You need to provide equivalence tests or Bayesian statistics to support these inferences (interpreting a null result in NHST as evidence for the absence of a difference or effect is not permitted). Please also address the concern regarding multiple testing in cases where a series of statistical tests applies to the same hypothesis.

I am attaching a checklist that details critical reporting requirements for the revised manuscript. Please attend to each item and ensure your manuscript is fully compliant. We are requesting that your manuscript aligns with these requirements as this facilitates the evaluation of your manuscript, reducing delays in re-review and potential future acceptance. If your revised manuscript is not aligned with these requests on major issues, such as those concerning statistics, it may be returned to you for further revisions without re-review. Additional information can be found in our style and formatting guide Communications Psychology formatting guide.

If the revision process takes significantly longer than five months, we will be happy to reconsider your paper at a later date, provided it still presents a significant contribution to the literature at that stage.

Please use the following link to submit your

- revised manuscript,
- point-by-point response to the referees' comments,
- cover letter (as a separate document),
- the Reporting Summary (see below), and
- the completed Editorial Request Table (attached):

Link Redacted

Thank you for the opportunity to review your work.

Best regards,

Marika Schiffer, on behalf of

Hannah Hao, PhD
Editorial Board Member
Communications Psychology
orcid.org/0000-0002-3342-9132

REVIEWER EXPERTISE:

Emotion regulation and psychophysiology

REVIEWER REPORTS:

Reviewer #1 (Remarks to the Author):

I appreciate the opportunity to review the manuscript "Laughter Regulation in Solitary and Social Contexts: Differential Effects of Suppression, Reappraisal, and Distraction." The authors examined the implications of three ER strategies for regulation the experience and expression of laughter. I applaud the authors for their efforts to self-replicate and for employing diverse measures, including both EMG and self-report data. Nonetheless, there are some major issues, particularly in conceptualization, which have limited my enthusiasm. I detailed them below.

Introduction:

Perhaps due to space limitations, the theoretical background was inadequate and inaccurate at times. First, many emotions are associated with laughter. For instance, people inferred different emotions based on different laughter sounds (Szameitat et al., 2009. Differentiation of emotions in laughter at the behavioral level. Emotion). Laughter is a multifaceted social behavior – which can differ in valence (taunting and schadenfreude vs. joy and tickling) – yet in the introduction, the authors did not describe how they defined laughter. This seems particularly relevant given that audio was used in the experimental stimuli.

On pg. 64, the authors wrote, "In contrast, our understanding of how individuals regulate positive emotions – particularly amusement – remains limited." I do not believe this is accurate. The authors seemed to have ignored the entire literature on the regulation of happiness. A lot of exceptional work has been on regulating pride, hope, etc, as well. Therefore, this statement is factually incorrect.

In studying the regulation of laughter, one key distinction needs to be made: are people regulating their emotions or expressions? These two types of regulations involve different regulation goals as well as different regulation strategies (Greenaway et al., 2021. Emotion experience and expression goals shape emotion regulation strategy choice. Emotion). For instance, expressive suppression is a strategy that people often choose when they want to experience but not express an emotion. Distraction is a strategy that people often choose when they want to neither experience nor express emotion. Goals dictate the selection of strategies in ER. These distinctions in goals are particularly relevant to understanding laughter, which involves both the regulation of experience and expression.

Related to this point, because the authors did not clarify whether they are focusing on the regulation of expression, the regulation of experience, or both, it becomes difficult to determine whether the focus of the three strategies are justified. Why studying these three strategies, but not others? The authors also did not explain why they expect the three strategies to yield different effects on "modulating facial expressions and perception of amusement." Moreover, if ER strategies are targeted at regulating expression/experiences, why would the authors expect the strategies to shape perceptions? Perceptions and experiences are related but conceptually distinct. There is a disconnect and swapping of the concepts here that needs to be justified

Also, it didn't occur to me that the authors were focusing on the down-regulation of laughter until the method section. This needs to be conceptualized as well. There are strategies to maintain or up-regulate laughter, such as rumination and

amplification. It seems problematic that these have not been considered at all.

Methods and results:

The authors said that sample size was determined based on prior study, how? What kind of power analysis was conducted? Was this preregistered?

Were there any kind of validation of the development of the stimuli, such as a pilot study?

PANAS measures were collected both before and after the manipulations, but the analysis of it was confusing to me. In the supplemental materials, the authors tested whether positive and negative ratings on this measure changed before and after the experimental manipulation. But did not account for condition? This seems odd. I'm also curious how much PANAS scores are correlated with ratings of funniness and why this affect measure wasn't considered an alternative DV that complement the single item of funniness.

Could there be order effects if block order was fixed? I'm also confused why strategy blocks were not randomized.

The authors only compared each regulation strategy to the control/unconstrained condition, but did not compare between them, why? If the goal is to understand the effectiveness of different strategies, why not compare the effects of the strategies? Similarly, in Experiment #3, the authors only reported the main effects but not the pairwise comparisons.

Related to my comments on the introduction, there was no clear justification provided for why distraction was added in Experiment 2. What contribution does including this condition make? The transition to Experiment 3 makes sense in terms of transitioning from a solitary context to including a social aspect. However, why does Experiment 3 only focused on suppression, but not the other two strategies - this is not justified.

Given the number of comparisons conducted, it seems pertinent to address Type I error and conduct statistical corrections.

Have the authors tried to include trial number as a predictor? Does the effects change across regulation trials?

In summary, this paper raises interesting questions, but I worry that the current studies are not well situated within the literature, and many design decisions are not well justified.

Reviewer #2 (Remarks to the Author):

There are many things to like in this manuscript: I am very appreciative of the rigorous methodology and the statistical analyses. The within-subjects designs are robust and sample sizes – well justified. Graphs provide an adequate visualisation of data and distributions. Below I list the points that should be addressed in the revision.

X First and foremost, the paper is currently organised around laughter. Yet, the central DVs in all three studies are an index of smiling (combination of Orbicularis Oculi, Zygomaticus Major, and Corrugator Supercilii EMG activity) and ratings of funniness. With these variables and in absence of other measures, there is simply no evidence that the paper examines laughter (rather than, say, smiling). I suggest reframing the paper to reflect the measures analysed.

X The introduction could bring a stronger case for the present studies. The authors are clearly very knowledgeable about the existing literature on emotion regulation (the real-world example with Angela Merkel worked wonders too), but the goal of the present research could be articulated more explicitly. In particular, I would like to see more detail about previous research on the use of emotion regulation in the context of positive emotion expressions. How do the present studies build on this existing research? What motivated the choice of regulation strategies? How do Studies 1 and 2 build on each other? What is the connection with Study 3? What is the role of funniness ratings? Of the PANAS scale? Some analyses (e.g., “Muscle inhibition modulated by social feedback”, p. 15; “Influence of humorous stimuli on mimicry suppression”, p. 16) appear a bit unexpectedly and would benefit from a more thorough introduction.

X When discussing predictions for each study, it would be helpful to specify the effects of interest (and, later, to connect these predictions – and effects – with the corresponding statistical analyses).

X I may have missed something – how were the jokes assigned to the three blocks in Studies 1 and 2? Were they randomly drawn from the initial pool? Was there any counterbalancing?

X The ratings of funniness seem low across the three studies with none of the means exceeding 3, again suggesting that smiles/Zygomatic activity could be a more relevant DV (unless other measures of laughter are available).

I hope that my comments will be useful and that the Authors will continue this interesting line of research.

EDITORIAL POLICIES

We ask that you ensure your manuscript complies with our editorial policies and reporting requirements.

To that end, we require revised manuscripts to be accompanied by a completed item: a reporting summary that collects information on study design and procedure.

- <https://www.nature.com/documents/nr-reporting-summary.pdf>>Nature Research Reporting Summary

Your revised manuscript can only be sent back to the referees if this checklist is completed and uploaded with the revision.

Notes: If you have submitted a Stage 1 Registered Report, Review, Primer, Comment, or Perspective you do not need to submit these forms. If you have already submitted these forms, you may disregard this request.

** Visit Nature Research's author and referees' website at <http://www.nature.com/authors>>www.nature.com/authors for information about policies, services and author benefits**

If you experience problems in linking your ORCID, please contact the <http://platformsupport.nature.com/>>Platform Support Helpdesk.

Version 1:

Decision Letter:

Dear Dr Mitschke,

Thank you for your patience during the peer-review process. Your manuscript titled "Laughter Regulation in Solitary and Social Contexts: Differential Effects of Suppression, Reappraisal, and Distraction" has now been seen by 1 reviewers, and I include their comments at the end of this message. They find your work of interest but raised some important points. We are interested in the possibility of publishing your study in Communications Psychology, but would like to consider your responses to these concerns and assess a revised manuscript before we make a final decision on publication.

We therefore invite you to revise and resubmit your manuscript, along with a point-by-point response to the reviewers. Please highlight all changes in the manuscript text file.

In line with Reviewer 2's comments, please focus the framing around amusement regulation, as opposed to laughter.

All preregistered hypotheses should be reported. At present, it is difficult to easily compare the preregistered hypotheses and their reporting in the manuscript. Please make this easier on the reader by specifically indicated the hypotheses numbers as included in the preregistration or including a table that maps the preregistrations onto the results reporting.

I am attaching an Editorial Requests Table that details critical reporting requirements for the revised manuscript. Please attend to each item and ensure your manuscript is fully compliant. If your revised manuscript is not aligned with these requests on major issues, such as those concerning statistics, it may be returned to you for further revisions without re-review.

Please submit the following items:

- Revised manuscript
- Point-by-point response to the referees' comments
- Cover letter (as a separate document)
- [Nature Research Reporting Summary](https://www.nature.com/documents/nr-reporting-summary.pdf)
- Completed Editorial Request Table (attached).

via this link: Link Redacted .

Additional guidance is available in our style and formatting guide [Communications Psychology formatting guide](https://www.nature.com/documents/commpsychol-style-formatting-guide-accept.pdf).

Best regards,

Hannah Hao

Hannah Hao, PhD
Editorial Board Member
Communications Psychology
orcid.org/0000-0002-3342-9132

REVIEWER EXPERTISE:

Reviewer #2 Emotion regulation and psychophysiology

REVIEWER REPORTS:

Reviewer #2 (Remarks to the Author):

Thank you for this responsive revision and engagement with both reviewers' comments.

While I am convinced by the new analyses, additional data, and the explanation of the choice of emotion regulation strategies, my key concern remains. The study assesses facial muscle activity and ratings of amusement. These are not laughter. Laughter involves vocalisations as well as bodily movement and contractions of the diaphragm and other abdominal muscles. Angela Merkel's laughter in the video mentioned in the introduction is signaled by a sound. Most laughter involves smiling, but not all smiling involves laughter. Moreover, smiles – just as well as laughter – can be used to express amusement. Therefore, the DVs of the paper should be explained as smiles/facial expressions of amusement and ratings of funniness, not laughter.

The Authors repeatedly mention the article by Korb et al. (2012) when explaining the sample size. While this article is about smiling, not laughter, supporting my earlier point, I think it is important to explain the methodology of this study and its relevance for the present research. Why is Korb et al. (2012) the best model to guide the sample size? And, more importantly, what key comparison/effect of interest was found in their research with 21 participants? This should be explained more clearly in the introduction and in the Participants section of Study 1.

O. 6: "listened to jokes solitary listening contexts" – should be: "listened to jokes in solitary..."

Version 2:

Decision Letter:

Dear Dr Mitschke,

Your manuscript titled "Laughter Regulation in Solitary and Social Contexts: Differential Effects of Suppression, Reappraisal, and Distraction" has now been editorially reviewed, and I am delighted to say that we are happy, in principle, to publish a suitably revised version in Communications Psychology.

We therefore invite you to revise your paper one last time to address the remaining editorial requests. At the same time we ask that you edit your manuscript to comply with our format requirements and to maximise the accessibility and therefore the impact of your work.

EDITORIAL REQUESTS:

SUBMISSION INFORMATION:

OPEN ACCESS:

* **DATA AVAILABILITY:**

Link Redacted

Best regards,

Jennifer Bellingtier

Jennifer Bellingtier, PhD
Senior Editor
Communications Psychology

Hannah Hao, PhD
Editorial Board Member
Communications Psychology
orcid.org/0000-0002-3342-9132

Dear Dr Schiffer, dear Dr Hao,

We would like to thank you for the constructive and detailed feedback on our manuscript titled "Laughter Regulation in Solitary and Social Contexts: Differential Effects of Suppression, Reappraisal, and Distraction." We are grateful for the opportunity to revise and strengthen our work.

In preparing this revision, we carefully addressed all reviewer comments, paying particular attention to the two editorially critical points outlined in your letter. All changes in the manuscript are highlighted in blue font for ease of review:

- **Methodological concerns regarding PANAS analysis and block order:** We clarified in the revised manuscript that the PANAS was administered only before and after the full experimental session (not per condition), and was therefore intended as an exploratory state check rather than a condition-sensitive outcome. This has been made explicit in both the Methods and Supplement. Regarding block order, we now justify our decision to use a fixed order in Experiments 1 and 2 and acknowledge the resulting limitations in the Discussion. We explain that this approach was chosen to avoid strategy spillover and ensure comparability across participants. We also clarify that the assignment of jokes to blocks was randomized for each participant.
- **Interpretation of non-significant results and multiple comparisons:** We have revised the Results and Discussion sections to avoid interpreting non-significant NHST results as evidence for the absence of an effect. Where relevant (e.g., for suppression effects on funniness ratings), we now include Bayes factors to quantify evidence for the null hypothesis. In addition, we have added a statement in the Statistical Analysis section regarding our approach to correcting for multiple comparisons when testing hypotheses across related outcome measures.

We hope these clarifications and additions adequately address the editorial concerns and demonstrate our commitment to methodological transparency and conceptual precision. Below, we provide a point-by-point response to all reviewer comments and outline the corresponding changes in the manuscript.

Sincerely,

Vanessa Mitschke (on behalf of all authors)

REVIEWER REPORTS:

Reviewer #1:

I appreciate the opportunity to review the manuscript "Laughter Regulation in Solitary and Social Contexts: Differential Effects of Suppression, Reappraisal, and Distraction." The authors examined the implications of three ER strategies for regulation the experience and expression of laughter. I applaud the authors for their efforts to self-replicate and for employing diverse measures, including both EMG and self-report data. Nonetheless, there are some major issues, particularly in conceptualization, which have limited my enthusiasm. I detailed them below.

Response: We sincerely thank Reviewer 1 for their thoughtful and detailed evaluation of our manuscript. We appreciate the careful attention to conceptual clarity and theoretical framing, and we found the suggestions highly valuable for sharpening the focus and rigor of the introduction and rationale. Below, we address each point in turn and describe the corresponding revisions made to the manuscript.

#1) Introduction: Perhaps due to space limitations, the theoretical background was inadequate and inaccurate at times. First, many emotions are associated with laughter. For instance, people inferred different emotions based on different laughter sounds (Szameitat et al., 2009). Differentiation of emotions in laughter at the behavioral level. Emotion). Laughter is a multifaceted social behavior – which can differ in valence (taunting and schadenfreude vs. joy and tickling) – yet in the introduction, **the authors did not describe how they defined laughter.** This seems particularly relevant given that audio was used in the experimental stimuli.

Response: We thank the reviewer for highlighting this important conceptual issue. We now provide a clearer definition and operationalization of laughter in the revised manuscript (abstract; Introduction - p. 2). Specifically, we emphasize that our study focuses on amusement-related laughter – a prototypical positive affect response – while acknowledging the broader emotional heterogeneity of laughter behaviors (e.g., Szameitat et al., 2009). To enhance conceptual precision, we clarify that our primary outcome measures (smile-related fEMG activity and funniness ratings) more accurately reflect the expression and experience of amusement, rather than vocalized laughter per se.

#2) On pg. 64, the authors wrote, "In contrast, our understanding of how individuals regulate positive emotions – particularly amusement – remains limited." I do not believe this is accurate. **The authors seemed to have ignored the entire literature on the regulation of happiness.** A lot of exceptional work has been on regulating pride, hope, etc, as well. Therefore, this statement is factually incorrect.

Response: We appreciate this important clarification and have revised the relevant passage in the introduction (p. 3) accordingly. We now explicitly acknowledge the growing body of research on the regulation of positive emotions such as happiness, pride, and hope (e.g.,

Quoidbach et al., 2014). At the same time, we emphasize that amusement – and its regulation in humorous contexts – has received comparatively less empirical attention, particularly in studies that combine subjective and physiological measures. Given the journal’s format constraints and the breadth of the literature, we opted to reference this line of research briefly rather than discuss it in detail. We hope the revised text strikes a balance between conceptual accuracy and conciseness.

#3) In studying the regulation of laughter, one key distinction needs to be made: **are people regulating their emotions or expressions?** These two types of regulations involve different regulation goals as well as different regulation strategies (Greenaway et al., 2021. Emotion experience and expression goals shape emotion regulation strategy choice. Emotion). For instance, expressive suppression is a strategy that people often choose when they want to experience but not express an emotion. Distraction is a strategy that people often choose when they want to neither experience nor express emotion. Goals dictate the selection of strategies in ER. These distinctions in goals are particularly relevant to understanding laughter, which involves both the regulation of experience and expression.

Response: This is an insightful observation, and we have addressed it directly in the revised introduction (p. 3). We now explicitly distinguish between expression- and experience-focused regulation, both conceptually and in terms of our outcome measures. Expressive suppression targets visible emotional behavior, distraction is primarily aimed at altering internal experience, and reappraisal may affect both. By analyzing both facial muscle activity and funniness ratings, we attempt to assess how each strategy modulates these distinct components of amusement. We do not claim to cover all aspects of laughter regulation but focus on tractable indices that reflect both expressive and experiential regulation outcomes.

#4) Related to this point, because the authors did not clarify whether they are focusing on the **regulation of expression, the regulation of experience, or both**, it becomes difficult to determine whether the focus of the three strategies are justified. **Why studying these three strategies, but not others? The authors also did not explain why they expect the three strategies to yield different effects on “modulating facial expressions and perception of amusement.”** Moreover, if ER strategies are targeted at regulating expression/experiences, why would the authors expect the strategies to shape perceptions? **Perceptions and experiences are related but conceptually distinct.** There is a disconnect and swapping of the concepts here that needs to be justified

Response: We thank the reviewer for pointing out the need to more clearly articulate the rationale for our strategy selection and hypotheses. In the revised manuscript (p. 4), we now explicitly justify our focus on expressive suppression, cognitive reappraisal, and distraction as theoretically grounded and commonly studied strategies that differ in their regulatory mechanisms and targets. These three approaches were chosen to represent a spectrum from expression-focused (suppression), to cognitively reinterpretive (reappraisal), to attention-diverting (distraction), each with plausible effects on both expression and experience.

We have clarified that our outcome measures assess both the *expression* (via facial EMG) and *subjective evaluation* (via funniness ratings) of amusement. While perception and experience are indeed distinct, we use funniness ratings as an index of *emotional appraisal*, reflecting the participant's evaluation of the humorous stimulus under different regulatory demands. We believe this framework allows us to examine how regulation strategies influence both overt expressive behavior and cognitive-affective judgments about humor, and we have revised the introduction accordingly to avoid terminological confusion.

#5) Also, it didn't occur to me that the authors were focusing on the **down-regulation of laughter until the method section**. This needs to be conceptualized as well. There are strategies to maintain or up-regulate laughter, such as rumination and amplification. It seems problematic that these have not been considered at all.

Response: We thank the reviewer for highlighting this important conceptual issue. We now provide a clearer definition and operationalization of laughter in the revised manuscript (p. 2). Specifically, we emphasize that our study focuses on *amusement-related laughter* – a prototypical positive affect response – while acknowledging the broader emotional heterogeneity of laughter behaviors (e.g., Szameitat et al., 2009). To enhance conceptual precision, we clarify that our primary outcome measures (smile-related fEMG activity and funniness ratings) more accurately reflect the *expression and experience of amusement*, rather than vocalized laughter per se. In addition, we have incorporated this point into the Introduction (p. 5) and Discussion (pp. 18, 24), where we explicitly acknowledge the exclusion of up-regulation strategies as a limitation and suggest that future research should explore amusement up-regulation goals and their role in promoting social bonding and shared affect.

#6) Methods and results: The authors said that **sample size was determined based on prior study**, how? What kind of power analysis was conducted? Was this **preregistered**?

Response: We thank the reviewer for noting the importance of a clear sample size rationale. As specified in our preregistrations (see Transparency and Openness, p. 25), the target sample size for Experiments 1 and 2 was based on Korb et al. (2012), who tested 21 participants but did not report a formal sample size calculation. To ensure adequate power and robustness, we oversampled to 40 participants. A sensitivity analysis for a repeated-measures ANOVA (assuming $\alpha = .05$, 80% power, and $r = 0.5$ for within-subject correlations) indicated that this sample size allows detection of effects of at least $f > 0.187$. We have now added this information – including the preregistration reference – directly to the Participants subsections of Experiments 1 and 2 for transparency.

For **Experiment 3**, due to the absence of directly comparable effect size estimates in the literature, we chose a target of 40 participants based on economic and feasibility considerations, with the intention (as preregistered) to conduct and report sensitivity analyses in the final manuscript.

#7) Were there any kind of **validation of the development of the stimuli**, such as a pilot study?

Reponse: We thank the reviewer for raising this important point. While we did not conduct a formal pre-validation or norming study prior to the current experiments, our stimulus selection was informed by several constraints and supported by independent data. We deliberately chose ethically appropriate, short verbal jokes that avoided offensive or discriminatory content, which limited the available stimulus pool. These jokes were then recorded by professional actors, making repeated pretesting infeasible due to production demands.

To address stimulus variability statistically, we treated joke ID as a random effect in all models, allowing us to account for trial-wise differences in perceived funniness within a mixed-effects framework. Additionally, subjective ratings in our experiments covered the full range of the funniness scale, indicating substantial individual variability.

To further support the suitability and variability of the stimuli, we now provide independent funniness rating data from a separate sample (N = 165) collected as part of a prior undergraduate thesis (Buchholz, 2023, supervised by the first author). These data confirm considerable item-level variability: funniness ratings ranged from 1 to 5, with a mean of 2.18 and SD of 1.20 across jokes.

Moreover, ratings of a similar magnitude are common in controlled laboratory humor studies. For example, Mayerhofer & Schacht (2015) reported mean humor ratings of 8.83 on a 3–15 scale (≈ 2.94 on a 1–5 scale) for humorous texts, Korb et al. (2014) found mean amusement ratings of 3.4 for humorous pictures and 2.58–2.62 when averaging across all trial types, and Ayçiçeği-Dinn et al. (2018) reported a mean rating of 3.2 (collapsed across Turkish and American samples) for humorous jokes. These values are highly comparable to our own and illustrate that moderate means are typical when humor is tested in controlled experimental settings rather than in naturalistic social contexts. (p. 23)

We hope this addition strengthens our rationale for using these materials without formal prevalidation and supports the robustness of our trial-level modeling approach.

#8) **PANAS** measures were collected both before and after the manipulations, but the analysis of it was confusing to me. In the supplemental materials, the authors tested whether positive and negative ratings on this measure changed before and after the experimental manipulation. But did not account for condition? This seems odd. I'm also curious how much PANAS scores are correlated with ratings of funniness and why this affect measure wasn't considered an alternative DV that complement the single item of funniness.

Reponse: We appreciate the reviewer's careful attention to the use of PANAS in our study. The PANAS was administered only once before and once after the experimental session, and not after individual blocks or conditions. As such, it does not allow for condition-specific

comparisons or correlations with trial-level funniness ratings. We have now clarified this in the revised Introduction and Methods section and in the supplement to avoid confusion.

Our goal in including PANAS was exploratory: to monitor general mood shifts across the session and ensure no unintended emotional consequences of the task. It was not intended as a core dependent variable. Instead, we chose funniness ratings as the main subjective outcome measure because they directly assess participants' evaluation of the jokes under different regulation strategies and social feedback conditions – precisely the focus of our hypotheses.

We agree, however, that future studies could benefit from more fine-grained, repeated affect measures (e.g., brief mood probes after each block) to complement funniness ratings and track broader changes in emotional state over time. We now briefly mention this possibility in the General Discussion (pp. 23-24).

#9) Could there be order effects if block order was fixed? I'm also confused why strategy blocks were not randomized.

Response: We thank the reviewer for raising this important point. The block order was deliberately fixed to begin with the unconstrained baseline, followed by expressive suppression, reappraisal, and (in Experiment 2) distraction. We now explain this rationale in the Methods section (p. 8). This fixed order was chosen to:

- Prevent carryover or contamination between regulation strategies (e.g., participants inadvertently continuing to reappraise or distract in later blocks),
- Reflect the cognitive complexity and natural application order of the strategies (i.e., suppression as a reactive default, followed by more effortful regulation), and
- Ensure participants first encountered the task without regulation instructions, promoting familiarity and interpretability of baseline responses.

These considerations were judged to outweigh the potential benefits of full counterbalancing, which could have introduced its own complications, such as strategy spill-over or diminished clarity of baseline responses.

To examine whether this fixed sequence might have systematically biased our results, we conducted exploratory mixed-effects models including trial number (z-scored per participant and condition) as a covariate.

In **Experiment 1**, the smile index showed a Condition × Trial interaction for suppression ($\beta = 0.138$, $SE = 0.06$, $t = 2.27$, $p = .025$): while smiles decreased across trials in the unconstrained condition, they increased slightly in suppression. For funniness ratings, the model revealed

two separate Condition \times Trial interactions – one for suppression ($\beta = -0.129$, $SE = 0.06$, $t = -2.22$, $p = .029$) and one for reappraisal ($\beta = -0.138$, $SE = 0.06$, $t = -2.39$, $p = .019$) – both indicating that ratings decreased across trials relative to the unconstrained condition.

These patterns suggest that temporal changes are not uniform across strategies: suppression shows a rebound in expression but a decline in perceived funniness, whereas reappraisal shows a decline in funniness without a rebound in expression.

Note. Standardized temporal dynamics of smile progression across time, divided by condition.

In **Experiment 2**, exploratory analyses revealed a significant three-way interaction of Condition \times Segment \times Trial for the smile index ($\beta = -0.118$, $SE = 0.03$, $t = -3.706$, $p = .001$). While smiles in the unconstrained condition weakened both across segments (within-joke time bins) and across trials, regulation conditions showed the opposite pattern – relatively stronger smiles later in the segment as trials progressed – indicating that temporal change was not uniform across conditions. In contrast, funniness ratings showed no main or interaction effects

of trial number, remaining stable across the session. This divergence between expressive and subjective measures, together with the fact that distraction (last in sequence) produced the strongest reduction in funniness ratings in the main analysis, is inconsistent with a simple fatigue or habituation explanation.

We acknowledge that this approach sacrifices full counterbalancing and have now included this as a limitation in the Discussion (pp. 23-24). Should the editor or reviewers find these exploratory analyses informative for readers, we would be happy to include them in the Supplementary Material.

#10) The authors only compared each regulation strategy to the control/unconstrained condition, **but did not compare between them, why?** If the goal is to understand the effectiveness of different strategies, why not compare the effects of the strategies? Similarly, in **Experiment #3, the authors only reported the main effects but not the pairwise comparisons.**

Response: We thank the reviewer for this valuable suggestion. Our initial analyses focused on comparisons with the unconstrained baseline because this provided the most direct test of whether each regulation strategy reduced amusement experience or expression relative to a natural, unregulated response. This approach aligned with our preregistered hypotheses, which were formulated as planned contrasts against the baseline.

We agree, however, that direct comparisons between strategies can provide important complementary insights into their relative effectiveness. We have therefore added the relevant pairwise comparisons to the Results sections for all experiments, where applicable. These allow readers to evaluate not only whether a strategy reduces amusement compared to the unconstrained condition, but also whether strategies differ from each other in their effects on subjective ratings and facial EMG.

#11) Related to my comments on the introduction, there was no clear justification provided for why **distraction** was added in Experiment 2. What contribution does including this condition make? The transition to Experiment 3 makes sense in terms of transitioning from a solitary context to including a social aspect. However, why does Experiment 3 only focused on suppression, but not the other two strategies - **this is not justified.**

Response: We thank the reviewer for pointing out the need to clarify the rationale behind our design decisions in Experiments 2 and 3. We have now expanded the Introduction (p. 4) and Methods section (p. X) to better justify the inclusion of distraction in Experiment 2. Specifically, distraction is a theoretically and functionally distinct regulation strategy that targets early attentional deployment rather than appraisal or expression (Gross & Thompson, 2007). Including distraction allowed us to test whether this relatively automatic and low-effort strategy would be more or less effective than reappraisal or suppression in modulating both amusement experience and expression.

Regarding Experiment 3, we now explain our decision to focus exclusively on expressive suppression (see p. X of the revised manuscript). Given the short temporal window and rapid demands of the laughter mimicry task, we selected suppression because it can be implemented quickly and reactively, making it suitable for testing in real-time interpersonal contexts. In contrast, distraction and reappraisal require more preparatory cognitive effort and are less compatible with a Go/No-Go-style design based on moment-to-moment facial mimicry. We now highlight this decision more clearly in the manuscript and discuss its implications as a limitation and future direction in the General Discussion (p. 24).

#12) Given the number of comparisons conducted, it seems pertinent to address Type I error and conduct statistical corrections.

Response: In the revised manuscript, we have replaced all reported p-values from post-hoc tests with values corrected for multiple testing using Bonferroni corrections. This correction controls the family-wise error rate while maintaining higher power than more conservative methods. All substantive conclusions remain unchanged after correction. However, since the preregistered hypothesis did not involve multiple tests per hypothesis, we have forgone an adjustment in these analyses.

#13) Have the authors tried to include **trial number as a predictor**? Does the effects change across regulation trials?

Response: We thank the reviewer for this thoughtful suggestion. As outlined in our response to Comment #9, we examined trial number as an additional predictor in our mixed-effects models for Experiments 1 and 2. The results showed condition-specific patterns rather than a uniform monotonic trend, indicating strategy-specific temporal dynamics rather than simple fatigue or habituation effects. Please see our detailed description in response #9.

#14) In summary, this paper raises interesting questions, but I worry that the current studies are **not well situated within the literature, and many design decisions are not well justified**.

Response: We thank the reviewer for this thoughtful overall assessment and greatly appreciate the opportunity to clarify and strengthen the theoretical grounding and methodological rationale of our work. In revising the manuscript, we have carefully addressed the points raised in the detailed comments above. Specifically, we have expanded the Introduction to better define our conceptual focus, more clearly justify our selection of regulation strategies, and specify the study's contribution to the emerging literature on amusement and laughter regulation. We also revised the Methods and Discussion sections to provide a clearer rationale for key design decisions – such as the choice of suppression in Experiment 3 and the fixed block order – and to reflect on their implications for future research.

We hope these revisions successfully communicate both the **novelty** and **rigor** of our approach and demonstrate how this work advances our understanding of how people regulate positive affect in socially dynamic settings.

Reviewer #2

There are many things to like in this manuscript: I am very appreciative of the rigorous methodology and the statistical analyses. The within-subjects designs are robust and sample sizes – well justified. Graphs provide an adequate visualisation of data and distributions. Below I list the points that should be addressed in the revision.

Response: We sincerely thank the reviewer for their thoughtful and constructive comments, as well as for their positive evaluation of the manuscript's methodological rigor, statistical analyses, and data visualisation. Below we address their suggestions for improving the conceptual framing and clarity of our manuscript.

#1) First and foremost, the paper is currently organised around laughter. Yet, the central DVs in all three studies are an **index of smiling** (combination of Orbicularis Oculi, Zygomaticus Major, and Corrugator Supercilii EMG activity) and ratings of funniness. With these variables and in absence of other measures, there is simply no evidence that the paper examines laughter (rather than, say, smiling). **I suggest reframing the paper to reflect the measures analysed.**

Response: We fully agree and have clarified this distinction both in the abstract and in the early part of the Introduction (p. 2). Specifically, we now define our focus as amusement-related laughter, elicited by humorous stimuli, and emphasize that our measures (fEMG activity and funniness ratings) serve as validated proxies for the expression and experience of amusement. We acknowledge that we do not capture vocalized or socially interactive laughter per se, and we have adjusted our terminology and framing to reflect this more precisely.

#2) The introduction could bring a stronger case for the present studies. The authors are clearly very knowledgeable about the existing literature on emotion regulation (the real-world example with Angela Merkel worked wonders too), **but the goal of the present research could be articulated more explicitly.** In particular, I would like to see more detail about previous research on the use of emotion regulation in the context of positive emotion expressions. How do the present studies build on this existing research? What motivated the choice of regulation strategies? **How do Studies 1 and 2 build on each other? What is the connection with Study 3? What is the role of funniness ratings? Of the PANAS scale? Some analyses (e.g., "Muscle inhibition modulated by social feedback", p. 15; "Influence of humorous stimuli on mimicry**

suppression", p. 16) appear a bit unexpectedly and would benefit from a more thorough introduction.

Response: We thank the reviewer for this helpful suggestion. We have made several changes to address these points:

- We now more clearly distinguish between expression-focused and experience-focused regulation, linking this distinction to our two outcome measures: facial EMG (expression) and funniness ratings (evaluation).
- We have elaborated the rationale for our choice of regulation strategies (suppression, reappraisal, distraction), emphasizing how they differ in their targets and mechanisms (p. 3–4).
- We added a short summary paragraph to clarify how the three experiments build on one another: Experiments 1 and 2 tested strategy effects in solitary contexts (with distraction added in Experiment 2), while Experiment 3 introduced social feedback to assess regulation success under interpersonal influence (p. 6).
- We clarified that PANAS was included as an exploratory measure of mood state, not as a core dependent variable (p. 4).

#3) When discussing predictions for each study, it would be helpful to **specify the effects of interest** (and, later, to connect these predictions – and effects – with the corresponding statistical analyses).

Response: We fully agree and have revised the Introduction (pp. 3–4) to clearly state our hypotheses for each strategy and each outcome. Specifically, we predict that suppression will primarily affect expression, reappraisal will influence both expression and experience, and distraction will dampen both. We also specify our expectations for the social context in Experiment 3, where we hypothesize that laughter feedback will increase facial mimicry and undermine suppression success. This framing more directly aligns the predictions with our theoretical goals and measured outcomes.

#4) I may have missed something – how were the jokes assigned to the three blocks in Studies 1 and 2? Were they randomly drawn from the initial pool? Was there any counterbalancing?

Response: We thank the reviewer for this important question. In both Experiments 1 and 2, jokes were randomly assigned to trials and presented only once per participant. In Experiment 1, this random assignment was implemented across all blocks, ensuring that each participant received a unique stimulus sequence. We have now clarified this in the Methods section (p. 6).

In Experiment 2, although the block order of regulation strategies was fixed, the assignment of jokes to conditions was also randomized for each participant, as in Experiment 1. We now make this explicit in the revised manuscript (p. X), and we acknowledge that while

randomization helped reduce systematic bias, a formal counterbalancing of stimulus-condition pairs across participants would further strengthen the design and is a valuable direction for future studies.

#5) The ratings of funniness seem low across the three studies with none of the means exceeding 3, again suggesting that smiles/Zygomatic activity could be a more relevant DV (unless other measures of laughter are available).

Response: We appreciate the reviewer's observation regarding the relatively low average funniness ratings across our studies. As noted, none of the condition means exceeded 3 on the 5-point scale. However, we believe this does not undermine the relevance of funniness ratings as a key dependent variable for several reasons.

First, funniness is a highly subjective and multifaceted construct that can vary widely across individuals, stimuli, and contexts. Importantly, our stimuli were selected to be ethically appropriate and broadly acceptable, which limited the inclusion of potentially more provocative or culturally specific humor that might have elicited higher ratings. Despite these constraints, participants used the full range of the funniness scale, and item-level variability was substantial, allowing us to meaningfully analyze both within- and between-participant effects.

Second, ratings of a similar magnitude are common in controlled laboratory humor studies. For example, Mayerhofer & Schacht (2015) reported mean humor ratings of 8.83 on a 3–15 scale (≈ 2.94 on a 1–5 scale) for humorous texts, Korb et al. (2012) found mean amusement ratings of 3.4 for humorous pictures and 2.58–2.62 when averaging across all trial types, and Ayçiçeği-Dinn et al. (2018) reported a mean rating of 3.2 (collapsed across Turkish and American samples) for humorous jokes. These values are highly comparable to our own and illustrate that moderate means are typical when humor is tested in controlled experimental settings rather than in naturalistic social contexts.

Third, the social modulation of funniness ratings (e.g., their reliable increase in response to laughter feedback in Experiment 3) supports their validity as a measure of amusement experience. This aligns with prior research demonstrating that perceived funniness is shaped by both internal appraisals and social cues.

Finally, we agree that smile-related fEMG activity is a highly sensitive and temporally precise index of emotional expression, and we therefore treated it as a complementary outcome throughout all three experiments. However, we maintain that subjective funniness ratings tap into the evaluative, cognitive-affective appraisal of humor, which may be modulated by emotion regulation strategies in ways that facial muscle activity alone cannot capture. (p. 23)

I hope that my comments will be useful and that the Authors will continue this interesting line of research.

Response: We sincerely thank the reviewer for their constructive feedback and encouraging closing remarks. We are grateful for the thoughtful engagement with our work and look forward to advancing this line of research further.

References:

- Ayçiçeği-Dinn, Ayşe, Şişman-Bal, Simge and Caldwell-Harris, Catherine L. (2018). Are jokes funnier in one's native language? *HUMOR*, 31, 5-37. <https://doi.org/10.1515/humor-2017-0112>
- Korb, S., Grandjean, D., Samson, A. C., Delplanque, S., & Scherer, K. R. (2012). Stop laughing! Humor perception with and without expressive suppression. *Social Neuroscience*, 7(5), 510–524. <https://doi.org/10.1080/17470919.2012.667573>
- Mayerhofer, B. & Schacht, A. (2015). From incoherence to mirth: neuro-cognitive processing of garden-path jokes. *Frontiers Psychology*, 6, 550. <https://doi.org/10.3389/fpsyg.2015.00550>

Dear Dr Hao,

We sincerely thank you and the reviewer for the constructive and encouraging feedback on our manuscript "*Laughter Regulation in Solitary and Social Contexts: Differential Effects of Suppression, Reappraisal, and Distraction.*"

We are pleased that the revised version was received positively and appreciate the opportunity to address the remaining points and editorial requests.

1. Conceptual framing: laughter vs amusement

We fully acknowledge the reviewer's concern and your editorial recommendation to clarify the conceptual distinction between laughter, smiling, and amusement. The overarching goal of our study was to test how distinct regulation strategies influence laughter responses to humorous stimuli, providing insight into mechanisms relevant for situations in which overt amusement must be controlled.

In our paradigm, participants in the suppression condition were instructed to *avoid showing how they felt* and to *relax their facial muscles so that no one could see what they were experiencing*. In the humorous task context, this instruction effectively required the inhibition of laughter-related expressions. The reappraisal and distraction instructions likewise targeted the reduction of amusement, either by reinterpreting the jokes or by shifting attention to an unrelated visual task. Together, these instructions represent distinct emotion-regulation mechanisms applied to amusement-eliciting material.

We therefore agree that our paradigm most accurately targets the regulation of amusement-related expressions and experiences, operationalized through facial muscle activity in the zygomaticus major and orbicularis oculi, which index the expressive component of laughter. This refinement aligns with your suggested phrasing ("the down-regulation of amusement-related expressions and experiences") and clarifies that the construct being regulated is amusement, as expressed through smiling and laughter. At the same time, the task's context made the behavioral goal of laughter inhibition transparent to participants, and our physiological measures capture the core muscular component of laughter expression—without activation of these muscles, laughter cannot occur. We have revised the manuscript accordingly to make this conceptual relationship explicit. The Introduction and Methods now clarify that (a) participants were instructed not to display their amusement while listening to jokes, (b) the three strategies represent distinct cognitive and expressive mechanisms for regulating amusement-related responses, and (c) facial EMG activity indexes the expressive manifestation of laughter.

In light of these clarifications, we respectfully suggest retaining the current title (*Laughter Regulation in Solitary and Social Contexts: Differential Effects of Suppression, Reappraisal, and Distraction*). The term *laughter regulation* most accurately reflects the behavioral instruction, the social-emotional context of the task, and the theoretical framework motivating the study. Moreover, the revised manuscript now explicitly situates laughter regulation as encompassing the regulation of amusement-related expressions and experiences, ensuring conceptual precision while maintaining terminological continuity with the preregistered design and previous versions of the manuscript. Should you nevertheless prefer a stricter focus on *amusement regulation*, we would be happy to adopt this change; however, we believe the current title best conveys the study's scope and contribution to research on emotion and expressive control.

2. Sample-size justification (reference to Korb et al., 2012)

We thank the reviewer for pointing out that our previous description of the sample-size rationale could be clarified. We now explain in both the Introduction and the Participants section of Experiment 1 why Korb et al. (2012) served as a methodological reference: their study used facial EMG to examine how emotion-regulation strategies modulate smiling responses to affective stimuli. Their within-subject comparison of suppression versus reappraisal yielded an effect size (Cohen's $d \approx 0.6$) that closely matches the magnitude of the effects we expected. Based on this precedent and our comparable design, we determined that a sample size of approximately 40 participants per experiment would provide adequate power (≈ 0.80) to detect medium-sized within-subject effects. The revised manuscript now summarizes these details and clarifies the relevance of the Korb et al. paradigm for our study.

3. Minor textual correction

We have corrected the typo on page 6 to read "listened to jokes **in** solitary listening contexts."

4. Minor Result Section adaptations

We have added CIs and additional statistics where requested. We have further removed the additional Post Hoc result section within the "Relation between experienced amusement and fEMG activity (Exp.1+2)" section since reporting the singular contrasts of funniness ratings within the model does not allow for a clear interpretation of effects and is redundant to the pre-registered analysis of ratings, which is already fully reported.

5. Summary of revisions

All requested changes have been implemented. Conceptual clarifications and the sample-size rationale were revised in the Introduction and Methods, while the typo correction was made on page 6. All changes are highlighted in blue font in the revised manuscript. The accompanying Editorial Request Table specifies where each requirement has been addressed.

We thank you again for your thoughtful guidance and for the opportunity to further refine and strengthen our work. We hope that these revisions address all remaining concerns and that the manuscript will now be suitable for publication in *Communications Psychology*.

With kind regards,

Vanessa Mitschke (on behalf of all authors)